# Strengthening the clinical academic pathway: a systematic review of interventions to support clinical academic careers for doctors and dentists

Gary Raine,[1] Connor Evans [ID],[1] Eleonora Petronella Uphoff,[1] Jennifer Valeska Elli Brown,[1,2] Paul E S Crampton [ID],[3] Amelia Kehoe,[3] Lesley Ann Stewart,[1] Gabrielle Maria Finn [ID],[4] Jessica Elizabeth Morgan [ID] [1,5]

¹Centre for Reviews and Dissemination, University of York, York, UK
²Mental Health and Addiction Research Group, Department of Health Sciences, University of York, York, UK
³Health Professions Education Unit, Hull York Medical School, York, UK
⁴School of Medical Sciences, Manchester University, Manchester, UK
⁵Department of Paediatric Haematology & Oncology, Leeds Children's Hospital, Leeds, UK

**Correspondence to**
Dr Jessica Elizabeth Morgan;
jess.morgan@york.ac.uk

## ABSTRACT

**Objective** Evaluate existing evidence on interventions intended to increase recruitment, retention and career progression within clinical academic (CA) careers, including a focus on addressing inequalities.

**Design** Systematic review.

**Data sources** Medline, Embase, Cochrane Controlled Register of Trials, PsycINFO and Education Resource Information Center searched October 2019.

**Study selection** Eligible studies included qualified doctors, dentists and/or those with a supervisory role. Outcomes were defined by studies and related to success rates of joining or continuing within a CA career.

**Data extraction and synthesis** Abstract screening was supported by machine learning software. Full-text screening was performed in duplicate, and study quality was assessed. Narrative synthesis of quantitative data was performed. Qualitative data were thematically analysed.

**Results** 148 studies examined interventions; of which 28 were included in the quantitative synthesis, 17 in the qualitative synthesis and 2 in both. Studies lacked methodological rigour and/or were hindered by incomplete reporting. Most were from North America. No study included in the syntheses evaluated interventions aimed at CA dentists.

Most quantitative evidence was from multifaceted training programmes. These may increase recruitment, but findings were less clear for retention and other outcomes. Qualitative studies reported benefits of supportive relationships, including peers and senior mentors. Protected time for research helped manage competing demands on CAs. Committed and experienced staff were seen as key facilitators of programme success. Respondents identified several other factors at a programme, organisational or national level which acted as facilitators or barriers to success. Few studies reported on the effects of interventions specific to women or minority groups.

**Conclusions** Existing research is limited by rigour and reporting. Better evaluation of future interventions, particularly those intended to address inequalities, is required. Within the limits of the evidence, comprehensive multifaceted programmes of training, including protected time, relational and support aspects, appear most successful in promoting CA careers.

## STRENGTHS AND LIMITATIONS OF THIS STUDY

⇒ This was a rigorous, systematic and transparent review conducted by a highly experienced research team.
⇒ Machine learning methodology facilitated very high-volume title and abstract screening to identify the most relevant records earlier than with traditional screening methods.
⇒ Limitations in the reporting of the existing literature made synthesis challenging.
⇒ It is unclear to what extent findings, which derived mostly from studies conducted in the USA, can be applied to other contexts. Multiple factors, including intercountry differences in organisational structures and practices, potentially limit the generalisability of findings.

**Systematic review registration** Open Science Framework: https://osf.io/mfy7a

## INTRODUCTION

Clinical academics (CAs), individuals who work in both clinical and research roles, are a key part of the academic and healthcare workforce, combining expertise from both roles for the benefit of patients. Their work is diverse, including any health professional background, and varying amounts of research and teaching commitments, dependent on individual's career stage, role and interests, as well as the healthcare, academic and wider social systems in which they operate. CA roles can bring benefits to the individual (through variety of work and career satisfaction), to patients (who benefit from high quality research and research active institutions) and to their institutions (through their transferrable skills, funding income and networks).[1–3]

The proportion of clinicians who choose a CA career has fallen over time.[4] Furthermore, there are inequalities within this career

path—with both gender and ethnicity differences being more pronounced than in either clinical or academic settings alone.[5][6] In 2017, less than 20% of UK CA professors were women, and less than 15% were of Black and Minority Ethnic (BME) backgrounds.[6] Thus, interventions are needed which both increase the numbers of CAs and facilitate increased equality of opportunity for those who wish to pursue this career.

A previous systematic review (searches up to 2017) summarised quantitative evaluations of interventions to improve gender equality in any academic discipline.[7] It found that the evidence base was restricted by poor quality, and that interventions were limited to approaches that required time and effort from the women they intended to support ('bottom-up' approaches).[7] Interventions to address other inequalities within the CA workforce have yet to be systematically explored and described. As such, there has been minimal synthesis of research into CA careers to date, and this has been limited in both methodological approach (quantitative studies only) and scope, though has produced interesting findings.

Career pathways for CAs frequently separate doctors and dentists from other members of the healthcare workforce. Therefore, our systematic review aimed to identify, critically appraise and synthesise research on existing interventions to increase recruitment, retention and career progression in CA medicine and dentistry. While we evaluated interventions focused on gender equality, we also sought interventions to address inequalities related to characteristics other than gender. Our research aims to inform regulators and funders of the most effective interventions to support and promote CA careers, with a view to increasing recruitment and retention within this group of healthcare professionals. CAs may also use our findings to negotiate their roles and advocate for support from those providing oversight to their careers.

## METHODS

The systematic review protocol was registered with Open Science Framework (https://osf.io/mfy7a) and published.[8]

### Search and information sources

The following databases were searched: Medline (including Medline Epub Ahead of Print, Medline In-Process & Other Non-Indexed Citations and Medline Daily), Embase, Cochrane Controlled Register of Trials, PsycINFO and Education Resource Information Center. Subject headings and free-text terms were used. Searches were limited to human studies published in English, from 2004 onwards (the introduction of the Athena SWAN initiative, a high-profile national programme aiming to improve gender equality across higher education).

We conducted two searches (see online supplemental appendix 1 for Medline strategy). One broad, sensitive search including terms relating to CAs. The other search was more specific and identified a subset of records from the broad search, by using terms relating to CAs and career development, recruitment, retention and attrition. Full details of the search process are provided in the protocol.[8]

### Inclusion and exclusion criteria

We included studies of qualified doctors, dentists and/or those with a supervisory role over these careers.[8] Studies of medical and dental students were not included. Studies of doctors and dentists who had completed their primary qualification but were undergoing further training (sometimes called specialty trainees, junior doctors, residents, fellows) were included, and are referred to as trainees within this manuscript. We included quantitative and qualitative studies evaluating interventions to increase recruitment to, and improve retention in, CA careers, using study-defined outcome measures relating to rates of joining or continuing within clinical academia. Conference abstracts were excluded. Studies were limited to those performed in high-income countries, according to the World Bank classification,[9] in recognition of the cultural and organisational setting in which the research findings are to be applied.

Given the large numbers of potentially relevant studies identified during the screening process, additional exclusion criteria were introduced at the full-text screening stage to focus on the most relevant evidence. We excluded studies published before 2005, studies where the majority of the data were collected before 2004 (again to reflect the post-Athena SWAN era), and studies conducted in high-income countries with considerable differences in culture and/or healthcare provision compared with the UK, for example Singapore. The analysis plan was adjusted to enable a time and resource efficient reporting process. Only quantitative studies with a control group and qualitative studies using data derived predominantly from verbal data collection methods were included in a detailed synthesis of data.

### Study selection and data extraction

We used the systematic review software Rayyan[10] to support the study selection process and employed our two-staged search process to train the built-in machine learning-based prioritisation function, with a view to more rapidly identifying potentially relevant records. Title and abstract screening took place in three stages, with all records from the most specific search and a randomly selected sample of the broader search screened in duplicate and used to train the Rayyan prioritisation function. The Rayyan prioritisation algorithm then supported the screening of the remaining records from the broader search. Screening stopped once the rate of potentially eligible records identified by the machine learning algorithm had fallen sufficiently from baseline.

Full-text screening was undertaken independently and in duplicate. At all stages, disagreements were resolved by a third reviewer or in team discussions.

Data were extracted by one researcher using standardised data extraction forms and independently checked by a second researcher. We extracted outcome data for only those studies that reported quantitative data and included a control group.

## Quality assessment

Quality assessment of studies included in the synthesis used the Cochrane risk of bias tool for Randomised Controlled Trials (RCTs),[11] the Newcastle-Ottawa scale for non-randomised studies,[12] the Qualitative Assessment and Review Instrument Checklist for qualitative studies,[13] the Mixed Methods Appraisal Tool for mixed methods studies[14] and the RAMESES (Realist And Meta-narrative Evidence Syntheses: Evolving Standards) II Quality Standards for Realist Evaluation.[15] Each study was individually assessed and checked by a second reviewer, and any conflicts resolved via discussion.

## Data synthesis

Data were summarised in narrative and tabular form. For accuracy of reporting, we retained the terminology used by included papers to describe their participants (which most often uses US terms). The frequently used American academic ranks of assistant professor and associate professor approximate those of lecturer and senior lecturer, respectively, in the UK.

Due to the heterogeneous nature of the studies, quantitative data were synthesised narratively, and qualitative data were synthesised based on the principles of thematic analysis.[16] All relevant qualitative findings were coded line by line by one researcher and codes subsequently reviewed by another researcher. Codes were developed inductively and further refined as appropriate. Findings related to specific codes were brought together to identify cross-cutting themes and issues of potential relevance.

## Patient and public involvement statement

Through Healthwatch York,[17] a member of the public was involved in the project steering group, influencing the inclusion criteria for the review to include international data, and informing the dissemination process.

## RESULTS
## Study selection

Electronic databases were searched in October 2019 and returned a total of 34 230 records; 148 studies examined interventions; of which 28 were included in the synthesis of quantitative data, 17 in the qualitative synthesis, and data from two studies were included in both (figure 1). The remaining 101 quantitative studies not including a control group were not synthesised further.

## Study characteristics
### Studies in the quantitative synthesis

Of the 30 studies included in the quantitative synthesis, 26 were conducted in the USA,[18–43] 2 in Canada,[44 45] 1 in Australia[46] and 1 in Germany.[47] Twenty-three were single

centre programmes,[18 19 21–24 26 28 29 31–43 47] and seven were national.[20 25 27 30 44–46]

No study included within the syntheses focused on CA dentists. Otherwise, the populations studied were varied, in terms of grade, academic level and medical background of participants (see online supplemental appendix 2). Due to this high degree of variability, it was often difficult to determine the exact population investigated. Fourteen studies focused solely on participants who had completed their medical training,[19–22 24 26 30 32–34 37 40–42] 11 studied just trainees,[18 23 25 28 29 31 35 43–45 47] 3 included mixed populations[27 36 38] and 2 were unclear/not reported.[39 46]

Studies encompassed a diverse range of interventions. The majority evaluated complex interventions involving elements such as mentoring, protected research time, leadership training and teaching workshops. Academic training programmes tended to focus on advancing trainee academic skills, research productivity and interest, while career development programmes (called faculty development programmes in some studies) centred on enhancing junior/senior faculty workforce within clinical academia through promotion, retention and recruitment.

Study design also varied. There was 1 RCT,[26] 2 case-control designs[25 40] and 27 studies with a cohort design[18 20–24 28–39 41–47] which included 2 studies using a mixed methods approach.[19 27]

Four interventions had a gender focus and were tailored specifically towards women,[20 22 26 42] two towards ethnicity/underrepresented minority faculty in medicine,[21 27] and two towards historically underrepresented faculty including women and minority populations.[24 41]

### Studies in the qualitative synthesis

Of the 19 studies included in the qualitative synthesis, 11 were conducted in the USA,[19 27 48–56] 5 in Canada[57–61] and 3 in the UK.[62–64] Eight were from single institutions,[19 53 55–57 59–61] and 11 were national level initiatives.[27 48–52 54 58 62–64] As in the quantitative synthesis, there was considerable diversity in the populations studied and the interventions involved (see online supplemental appendix 2). Ten of the included studies used qualitative methodology only,[49–52 54 56 57 59–61] eight studies used a mixed/multiple methods approach[19 27 48 53 55 58 63 64] and one study was described as a realist evaluation.[62]

Two interventions were aimed at women,[51 54] one at individuals from ethnic groups under-represented in medicine[27] and another at 'busy clinician educators'.[55] One study reported on an intervention targeted at participants who experienced substantial caregiving challenges.[52]

### Quality assessment

Both quantitative and qualitative studies lacked methodological rigour and/or were hindered by incomplete reporting. Across all study types, intervention characteristics and population definitions were reported ambiguously. In most cohort studies, there was minimal participant matching between intervention and control groups, and participant comparisons were often unadjusted or

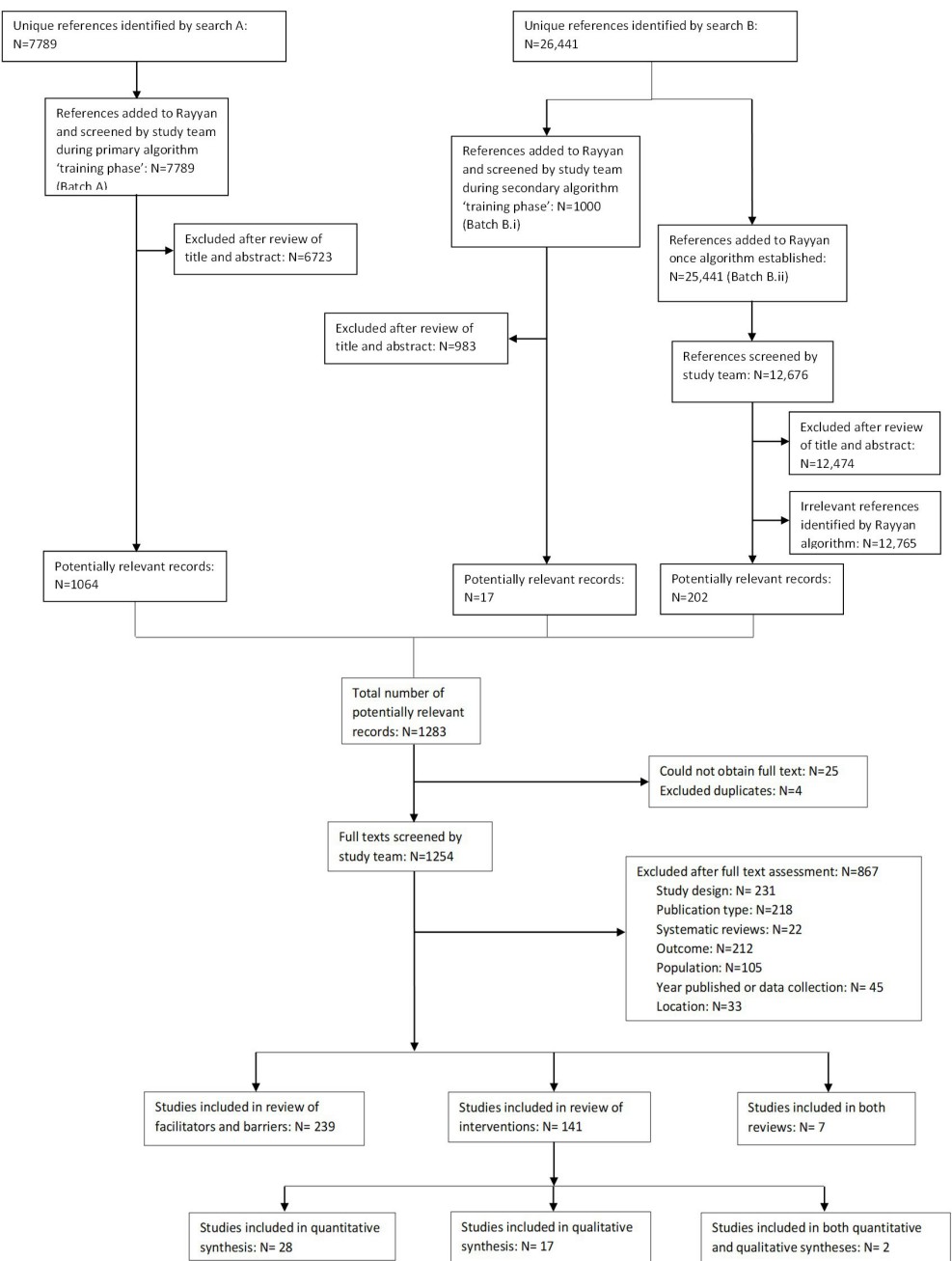

**Figure 1** Flowsheet for study selection.

ill-defined. Group selection was equally problematic as many studies included preselected or highly motivated populations to receive the interventions, which may unintentionally bias results in favour of the intervention programmes. Nonetheless, the quantitative literature generally included appropriate follow-up methods and a large volume of outcome data suitable for analysis. The qualitative studies also provided rich data from relevant populations. However, comprehensive analysis plans and complete methodological reporting were not provided in most qualitative studies, and there was little reflexivity demonstrated. For further details on quality assessments, see online supplemental appendix 3.

### Synthesis of quantitative data

All relevant reported outcomes were extracted and then grouped under eight broad categories relating to CA careers (with categories created after the identification of outcome data). These categories were: aspiration, career satisfaction, skills and knowledge, research funding, research participation, recruitment, retention/promotion and publication outcomes (full outcome data are provided in online supplemental appendix 4).

### Aspiration

Only one study included a measure of aspiration. It found that significantly more participants attending a career development programme aspired to achieve a higher

leadership position in academic medicine compared with non-attendees.[22]

## Career satisfaction

Three studies reported outcomes relating to career satisfaction.[19 26 43] No single outcome demonstrated a statistically significant benefit for intervention participants versus controls, although results were favourable towards the intervention participants.

## Skills and knowledge

Four studies reported outcomes relating to skills and knowledge.[19 31 43 47] Significantly higher clinical competence scores,[31] and improved research competence scores[47] were found among participants of two academic training programmes. Methodological research knowledge also significantly improved for intervention participants in one study.[47] Significantly more intervention participants learnt how to give a presentation in the study by Winn et al[43]; yet, within the same study there was no significant difference in the percentage of participants who learnt how to present a poster or felt prepared for scholarly work after postgraduate training, compared with controls. No significant benefits were found for the Academy for Collaborative Innovation and Transformation career development programme.[19]

## Research funding

Outcomes relating to research funding were reported in six studies.[25–27 33 46 47] Three studies found significant increases in the number of funded grants[27 33] or the percentage of people with successful grant applications[47] for participants in programmes which combined research training with mentorship. Similar findings were non-significant in two training and development programmes.[25 26] Another study found an increase in number of grant awards for intervention participants but it is unclear if this difference was statistically significant.[46] Two out of three studies measuring the amount of funding received (both amount of money and number of grants) found significantly greater amounts of funding awarded to participants attending career development programmes with mentorship compared with non-attendees.[27 33]

## Research participation

Research participation outcomes were reported in eight studies.[18 25 26 28 32 34 46 47] Four studies included a measure of involvement in research activities[18 25 32 46]; however, only two studies showed a significant increase in research participation—both studies assessed multifaceted academic training programmes.[18 25] Mandel et al[34] found similar benefits for graduates taking part in a research-tailored curriculum intervention but it is unclear if this finding was significant. Two out of three measures relating to grant applications did not show any difference between groups in two interventions that included comprehensive research training.[28 47] Löwe et al[47] also found no benefit

of a training programme on the number of participants writing a 'book article' during residency.

## Recruitment

Nine studies reported outcomes relating to recruitment in academia and achievement of a specific academic rank.[18 22 24 25 29 30 41 42 46] Two studies found that participants attending academic training programmes were significantly more likely to obtain their first job in academic practice post training compared with non-attendees.[18 29] Three studies showed that intervention participants were significantly more likely to be recruited at higher academic ranks,[22] and achieve higher ranks of assistant professor[25] or full professor[30] more quickly than participants not enrolled in academic training programmes. Female faculty recruitment increased in two intervention studies aimed at improving diversity.[41 42] Valantine et al[42] found significant increases in recruitment at full professor level but no significant increase was evident at assistant or associate professor level. Similarly, Emans et al[24] found that female faculty recruitment for intervention participants only significantly increased at associate professor level opposed to non-significant increases for other academic levels.

## Retention and promotion

Ten studies reported outcomes relating to retention, or promotion to a higher position.[20 21 23 24 27 31 32 37 39 40] The retention of staff in clinical academia was significantly higher for participants in two career development programmes[39 40] but non-significant in another career development programme,[27] and two academic training programmes.[23 32] In one study, residents of an academic training programme with research experience were significantly more likely to choose academia as a future career than participants in the control group.[32] One study found significantly higher retention rates for assistant professors who took part in a career development programme tailored for women,[20] while a similar programme aiming to improve diversity found no difference in retention of underrepresented minority staff.[21] One mentorship programme found significantly higher rates of promotion to associate professor in the intervention group,[37] while another study found no increase in promotions for participants in the intervention group.[27]

## Publications

Publication outcomes were reported in 14 studies,[18 23 25–29 32 35 36 38 44 45 47] all of which measured the number of publications during or within 5 years of receiving an intervention. Nine studies found a significant increase in publication productivity, of which, eight were academic training programmes[18 23 25 29 32 35 45 47] and one was a mentorship programme.[36] The other five studies found either non-significant differences[26–28 44] or differences with no statistical analyses reported.[38] The number of first-author publications was reported in five studies[23 26 29 45 47]; but only three studies, all evaluating

an academic training programme, showed a significant increase for intervention participants.[23 29 45] Publication impact was also measured across four studies, but only two studies demonstrated significantly higher H-index scores[23] or impact factors scores[29] for intervention participants taking part in an academic training programme. The number of peer review journal articles[26] and books published[47] were not found to be significantly different between intervention and control groups.

### Synthesis of qualitative data

We identified seven themes:

### Developing knowledge, skills and confidence in research and scholarship

Participants in seven studies reported improvements in research/scholarship knowledge and skills, and their confidence to conduct research activities, from planned teaching or learning sessions.[19 48 51 55 57 59 61] Respondents across three studies mentioned specific programme components as being useful for developing research/scholarship knowledge and skills.[48 55 61] The didactic sessions considered most helpful in one study focused on literature reviews, survey methodology and reference management.[55] Participants in another programme reported an unmet need for skills development including negotiation, grant management and work–life balance, alongside a need for advice about a lack of faculty diversity and unconscious bias.[27]

### Leadership skills and opportunities

Four studies reported positive findings about the leadership skills or opportunities that participants gained.[19 27 51 54] Respondents in one study gained leadership experience with a women-focused national organisation at an earlier career stage than anticipated, and this helped some earn promotion.[54] Individuals in two studies pursued new leadership opportunities, or progressed their careers in other ways, after gaining self-confidence in career development programmes.[19 51]

### Personal characteristics and behaviour of individuals

Personal characteristics, including participants' level of personal ambition, enthusiasm, motivation, self-direction, interest and commitment to the programme, influenced the decision to initially apply to a programme and/or participants' programme experiences in six studies.[49 51 56 60 63 64] Some participants in the UK Academy of Medical Sciences mentoring scheme reported less successful relationships with mentors due, at least in part, to mentees not having been sufficiently proactive in arranging meetings and maintaining contact with mentors.[64] In the UK Academic Foundation Programme, good preparation by trainees facilitated success, including arranging early contact with supervisors and maintaining engagement at key stages of training. Conversely, one supervisor thought a barrier to success was that some candidates did not appreciate the time and work involved.[63]

### Interactions and relationships

Networking through participation in career development initiatives, especially with individuals from other institutions, was valued for the benefits to career advancement and opportunities to identify mentors, and develop relationships with peers.[27 51 54] One study indicated that sponsorship was of benefit to women in terms of career advancement, including nominations for promotion and/or writing supportive references.[54]

Intervention participants gained emotional support, encouragement, self-confidence and other benefits from relationships with peers in 11 studies.[19 27 48 49 51 54–57 61 63] Interacting and engaging with peers reduced feelings of professional isolation[51 54 56 61] and fostered a sense of community and belonging.[19 54 61] Women valued peer interaction, and the opportunity to share experiences, with other women.[49 51 54] Respondents also appreciated the professional collaboration that arose from peer relationships.[19 48 49 51 54 57 63] Some findings related to interactions explicitly described as peer mentoring.[19 27 49 55 61] Notably, we identified multiple terms used to describe peer interaction which made it difficult to draw clear distinctions between such terms and how they represent different forms of peer support.

Participants spoke positively about mentorship from senior colleagues.[27 48–51 54–57 59 60 64] In addition to being a source of moral support, self-confidence and encouragement, mentors provided other assistance, including offering career-related advice; teaching research and scholarship skills; facilitating leadership opportunities; assisting with grant applications; suggesting ways of dealing with rejection and setbacks; and providing resources such as staff or equipment.[27 48–50 57 64]

> When the challenges of combining clinical training and research clouded my judgment about future career steps, my mentor proved to be indispensable in making the most objective and adequate choice.[64]

Respondents believed it was important to develop a 'network' or team of several mentors, drawing on the different strengths and areas of expertise of each mentor, and guarding against inadequate mentoring.[48 49 54] Having at least one female mentor was important to some women in four studies, particularly in terms of having a role model, and providing guidance on balancing a career and family life.[49–51 54] One study found mixed views on whether both mentor and mentee need to be from an ethnic group under-represented in medicine.[27] A respondent in another study believed some individuals may experience difficulties in finding mentors of the same ethnicity.[49] Mentees in two studies believed they gained more objective and impartial advice from having mentors who were from a different institution.[49 64] Trainees in one study preferred physicians over non-physicians as mentors, viewing them as role models.[60]

Studies reported mixed findings on the benefits of formalised mentorship programmes,[60 64] and on the need for training of mentors and mentees.[27 48 63 64]

## Time and competing demands in clinical academia

Issues related to the time pressures experienced by CAs formed a consistent narrative across studies. Protected time was an important feature of career development programmes and training fellowships in seven studies.[19 48 52 55 57 58 60]

> Really what I needed was dedicated time so I'd have relief time from clinic to work on the project…it gave me a chance to… really move the project forward a lot more than I would have without it.[55]

One funding award enabled participants with substantial care-giving demands to gain greater control, flexibility, and choice over their time through buying more protected research time and hiring staff to take over various research-related tasks.[52] Awardees reported greater research productivity and an improved work–life balance. The award also assisted with career progression and retention in academia at critical time points. In another study, a participant found protected time facilitated achieving more publications.[48]

Some participants found it difficult in practice to maintain dedicated research time due to lack of clinical cover, and found that administrative staff were not always supportive which made scheduling protected time difficult.[52 55] Time conflicts, particularly competing clinical demands, acted as a barrier to organising or participating in specific programme elements, research training or implementing training fellowships across four studies.[52 55 59 60] The impact of the Athena SWAN programme was potentially undermined by institutions holding meetings at times that could be difficult for staff with caring responsibilities to attend.[62]

The competing commitments of clinical supervisors were barriers to success in the UK Academic Foundation Programme,[63] while in Canada, several Clinician Investigator Programme directors did not have protected time for managing the programme.[58] Similarly, competing demands on senior faculty staff detracted from their ability to be a good mentor in another study.[49]

## Facilitating programme participation and success

The influence of programme and organisational level factors on intervention participation and impact was identified across studies.

Having committed and experienced programme staff facilitated success in six studies.[19 48 55 57 58 60] We interpreted 'programme staff' as individuals involved in programme delivery, including teachers, administrative staff and programme leads.

Management support influenced programme participation in four studies.[19 51 58 62] In one study, the provision of financial support to women faculty had enabled them to attend a career development programme.[51] In contrast, postdoctoral researchers perceived a lack of support from project leads and expressed scepticism that they would be given time to participate in Athena SWAN activities.[62]

In one study, most respondents supported the Athena SWAN programme and believed it had positive institutional outcomes.[62] However, the application process increased the workload of the mostly female self-assessment team, reinforcing institutional gender inequity and undermining the programme's aims.[62] A perceived emphasis on only supporting women was considered to perpetuate existing gendered social norms related to childcare provision.[62]

Two studies identified issues related to the promotion of programmes to staff.[51 62] Participants had not accessed initiatives, as they were unaware of them or had ignored relevant information to prevent 'email overload'.[62] Elsewhere, a lack of institutional information promoting programmes meant faculty only learnt about them from colleagues.[51]

Seven studies identified factors influencing success related to the delivery of learning sessions and training fellowships.[19 48 58–61 63] Involvement in planning and developing sessions was considered important.[61] Various teaching methods were influential including: diverse educational methods; mixed guided and independent learning approaches; experiential learning; tailored coursework; and using the expertise of staff from different institutions.[19 48 59 60] Other beneficial aspects of programmes included: a 'supportive' learning environment, high degree of autonomy in research training, flexibility, structure and clear guidance.[58 59] Barriers included some didactic content being judged as too jargonistic and delivered using inaccessible language.[19] Supervisors in the UK Academic Foundation Programme considered the short period of academic time a barrier.[63] Challenges related to infrastructure and logistics, for example the lack of suitable desk space, were found in two programmes.[56 60]

At a national level, UK funding arrangements could undermine family-friendly policies implemented to support Athena SWAN, for example by not providing funding for maternity cover in grant awards.[62] Meanwhile, in Canada, inadequate funding for trainees was viewed as a barrier to Clinician Investigator Program entry.[58]

## Funding and financial support

Funding for protected time and issues related to national policies are discussed earlier. Two studies suggested improvements in the intervention including funding administrative staff or research personnel[52 55] and one reported that lack of funding resulted in limited access to specialist statistical support.[63] Funded fellowship programmes were reported to be more successful because they provided protected time for developing programme infrastructure.[60]

The need for bridge or seed funding was mentioned in some studies.[52] Funding requirements considering time worked rather than chronological time limits were also deemed important:

[A]pplications [where] eligibility [is limited to] 3 years within starting your faculty position or 8 years within graduating [should] have this prorated…. so that the eligibility is based on time worked, not just a chronologic year, which may have a 3-month maternity leave… in it.[52]

Mentorship could help address some of the financial challenges for junior CAs, through supporting grant application writing, helping with bridging funds or providing research and administrative staff support.[50]

## DISCUSSION

### Statement of principal findings

We identified few high quality, well-reported evaluations of interventions to improve recruitment or retention to clinical academia. Most studies were from North America and no controlled quantitative studies were from the UK. No studies included interventions for CA dentists, and few included specific interventions for women or minority groups. No studies reported on outcomes related to patient benefit, or cost-effectiveness of interventions.

Most quantitative evidence derived from multifaceted academic training programmes; such programmes may increase recruitment to academia among clinicians, but findings were less clear for retention or other outcomes related to participation in research and research funding. Qualitative studies reported benefits of supportive relationships for CAs, including peer and senior mentors. Formalised mentoring programmes were not universally considered useful. There was consistent evidence of the importance of having protected time, particularly to mitigate against the negative impact of competing clinical demands on research-related activity, though maintaining protected time could be difficult in practice. Across studies, committed and experienced programme staff were key facilitators of success. This study adds to the existing literature by detailing more information about the evidence base of a broad range of interventions to increase recruitment and retention to CA careers, highlighting a clear gap in addressing inequalities.

### Strengths and limitations of this review

This review used rigorous, systematic, and transparent methods conducted by a highly experienced research team. The broad focus provides insights for CAs, programme leads and funders. We used machine learning methodology to facilitate very high-volume title and abstract screening to identify the most relevant records earlier, while resulting in better resource management by reducing the screening burden for the research team and reducing the time to review completion. As experience with machine learning grows, this may be further improved as confidence in the algorithms increases. We may have missed a small number of potentially relevant records (<1% of the total), which we do not anticipate to have impacted on the overall review results. Nonetheless, the high volume of potentially eligible records and limited time resource meant it was not possible to investigate all relevant primary studies in depth.

Limitations in the reporting of the existing literature made synthesis difficult. Many studies involved highly motivated participants compared with controls or participants who did not get accepted onto a programme as control groups. Such groups are not directly comparable leading to potentially biased results caused by baseline differences. Furthermore, it is unclear to what extent findings, which derived mostly from studies conducted in the USA, can be applied to other contexts. Multiple factors, including intercountry differences in organisational structures and practices, potentially limit the generalisability of findings. These challenges in extrapolating from literature limited by quality and geography hindered our ability to draw robust conclusions on the effectiveness of interventions designed to support CA careers within the UK.

### Our findings in relation to previous studies

This review includes a more substantial volume of literature than the previous review by Laver et al.[7] Our work confirms their findings that most studies are from the USA and that the quality of research in this field is generally poor. Compared with the previous systematic review, our review used a more comprehensive search strategy, and included interventions for all CAs, not just women. Our review had a broader focus than solely interventions to promote gender equality, but also found most evidence supported multifaceted interventions and those with a mentoring component.

Our systematic review rigorously synthesises a previously disjointed body of evidence from a wide variety of methodologies and sources, thus presenting a coherent summary of the state of research landscape in this area. This goes beyond the previously available evidence about inequalities in CA careers which often relied on small single centre studies, personal accounts or routinely collected data. Policy and decision-makers will be able to use our systematic review with confidence when planning and implementing future interventions.

### Implications of this review for clinical academia and policy-makers

The key implication identified by this review is that multifaceted interventions are most likely to be successful in promoting CA careers, with components such as protected time, mentorship and supportive staff. The contribution of each facet of the intervention was not always clear due to incomplete reporting, and thus future work may consider using methods such as realist evaluation to distinguish the role each component plays in a programme's ability to deliver key outcomes for CAs.

The lack of quantitative evidence of benefit should be carefully considered when planning future mentorship interventions, particularly the need for detailed evaluations. While our qualitative synthesis showed mentoring

of junior CAs by seniors was often considered beneficial, the quantitative data were less supportive of mentoring interventions, with some findings in favour, but others finding no evidence of benefit. Most evidence suggested mentorship improved funding received and supported programmes incorporating both peer and senior mentorship. Mentorship team composition may play a key role in the success of the relationship, including factors such as the number, gender and location of mentors. Little evidence related to mentor ethnicity. Formalised mentorship schemes, including requiring specific mentor characteristics, were not clearly supported by the literature. Our findings may reflect the design of individual programmes, or the studies evaluating them, but the need for further evaluation of mentorship programmes is clear.

Some studies suggested personal attributes of individual participants, such as commitment, enthusiasm and motivation, could be key influences on intervention success, thus placing responsibility for programme failures on the individuals involved, rather than institutional or cultural factors. The complex interplay of gender, ethnicity, parenthood and other protected characteristics on these attributes has not been explored within the included literature. Indeed, there was a notable absence of intersectional focus within the qualitative literature overall, with just two studies highlighting the interactions between gender and ethnicity.[52 54] In the quantitative synthesis, few studies focused on interventions tailored towards women and/or underrepresented minority faculty, and none evaluated an intervention from an intersectional standpoint. Adopting an intersectionality perspective when developing and evaluating future strategies may address more effectively issues related to inequality within clinical academia.

Various factors at organisational or national levels had negative impacts on the success of interventions within this review. Such evidence indicates that the success of future initiatives will be limited unless action is taken to ensure that organisational practice and culture, as well as relevant national policies, support the recruitment, retention and progression of CAs.

### Implications of this review for research

From the identified evidence on interventions, it is clear that little benefit will be derived from conducting further small, single-centre cohort studies in this field. Future research should use more robust methods to evaluate the effectiveness of interventions over time, using a control group and outcomes relating to recruitment and retention for medical and dental academics. We recommend that research funders commit to establishing large scale national research infrastructure to facilitate this, spanning both clinical and academic environments.

Interventions evaluated through this infrastructure are most likely to be successful when embedded within comprehensive multifaceted programmes, focused on developing relationships between CAs. Consideration should be given to evaluating support structures, including administration, personnel and programme leadership, as well as the most effective timing of interventions along the CA career path. Evaluations of structural and environmental factors should be prioritised over interventions targeting individual determinants.

Clear terminology for describing CAs, and coproduced core outcome sets for studies of CA careers, would aid the synthesis of primary studies in future systematic reviews. To allow for a focus on equality and diversity, results should be open-access and transparently presented in disaggregated form, reporting gender and ethnicity differences. In addition, the intersectional perspective is notably absent from this field and further high-quality, reflexive, qualitative research should explore the interplay of multiple determinants of inequality.

Within this review, we identified an additional dataset of studies evaluating barriers and facilitators to CA careers. Further exploration and synthesis of this dataset may facilitate a deeper understanding of the barriers and facilitators that CAs face and may inform the development of specifically targeted interventions. This systematic review formed part of a larger project focused on UK CA careers, for which linked primary qualitative research was performed.[65 66] A full and detailed report of the whole project is available online.[67 68]

**Contributors** JEM, GMF, PESC and LAS designed the study in collaboration and obtained funding as detailed. Search strategies were developed, tested and translated by an information specialist with input from JVEB and JEM. JVEB and JEM drafted the protocol and then revised it alongside the other authors. JVEB, JEM, CE, GR, EPU, PESC, AK and GMF screened titles and abstracts for inclusion. JEM, JVEB, CE, GR and EPU screened full texts, performed data extraction, and quality assessment. JEM, CE, GR and EPU performed the analyses. All authors were involved in the writing and editing of the manuscript. All authors have read and approved the final manuscript. JEM is the guarantor for the mansuscript.

**Funding** This systematic review, as well as the larger project it is part of, was funded jointly by the Medical Research Council (MRC), the National Institute for Health Research (NIHR), Wellcome, Health Education England (HEE), the Academy of Medical Sciences, and Cancer Research UK (CRUK) through an award administered by CRUK [C71037/A29824]. The funders identified the focus of the work and the use of systematic review methodology to address the research question. The protocol for and findings of the study were not further influenced by the funders.

**Competing interests** None declared.

**Patient and public involvement** Patients and/or the public were involved in the design, or conduct, or reporting, or dissemination plans of this research. Refer to the Methods section for further details.

**Patient consent for publication** Not applicable.

**Ethics approval** Not applicable.

**Provenance and peer review** Not commissioned; externally peer reviewed.

**Data availability statement** Data sharing not applicable as no datasets generated and/or analysed for this study. Data included within this review have been extracted from the included studies, in their published form. No additional data are available.

for any error and/or omissions arising from translation and adaptation or otherwise.

**ORCID iDs**
Connor Evans http://orcid.org/0000-0002-4525-2100
Paul E S Crampton http://orcid.org/0000-0001-8744-930X
Gabrielle Maria Finn http://orcid.org/0000-0002-0419-694X
Jessica Elizabeth Morgan http://orcid.org/0000-0001-8087-8638

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
