## [Reviewer comments · BMJ Open]

ARTICLE DETAILS

TITLE (PROVISIONAL)	Strengthening the clinical academic pathway: a systematic review of interventions to support clinical academic careers for doctors and dentists
AUTHORS	Raine, Gary; Evans, Connor; Uphoff, Eleonora; Brown, Jennifer; Crampton, Paul; Kehoe, Amelia; Stewart, Lesley; Finn, Gabrielle; Morgan, Jessica

VERSION 1 – REVIEW

REVIEWER	Sue Latter University of Southampton
REVIEW RETURNED	08-Feb-2022

GENERAL COMMENTS	On the whole this is a well conducted review with clearly presented results and subsequent discussion and implications that follow from these. The critical details missing are on the details of the search process, and also the justification for and details about the decision to exclude 101 of the 148 studies in the interests of a feasible review. Specific comments: Abstract: Objectives: is it 'a focus on addressing inequalities'? This did not seem the prime focus of the review. Perhaps better as 'including a focus on addressing inequalities'. Is there a qualitative equivalent to 'risk of bias' that was applied, and therefore should be included in the Abstract, for balance? Results: What is the 'general overview'? This is unclear, and is not reported in the body of the text either. Strengths and limitations: 'Identify...earlier' is a little unclear. Earlier than what? Introduction: Second paragraph: could the relevance of this be made more explicit? Interventions designed to increase career progression is stated as an aim – if the latter distinct from recruitment and retention, should it be included in the Abstract and the body of the text as a focus? Is it 'a key group' or group of people plural wrt inequalities? Methods It would be helpful to spell out in full which database the review was
--

	registered with. The search strategy seems somewhat unconventional and the authors says the details will be published separately, but sufficient details should be included in this paper. No inclusion / exclusion criteria related to inequalities? Or career progression? Data are plural. Why was the inclusion date changed from 2004 to 2005? Assume this was to allow for the time lag between Athena Swan introduction in 2004 and any research conducted in its wake? Please clarify. The analysis plan was adjusted? Further detail and a rationale for this are required. Study selection: The rationale for excluding 101 studies needs to be included. Page 6: trainees were included or not included? Exclusion criteria states students were not included – if these are different than ‘ trainees’ a word of explanation is needed. It would be helpful to summarise the range and types of interventions included. Were interventions included if they promoted careers generally as opposed to CA careers specifically? It would be helpful to summarise the range and types of design of studies. Can the authors comment on the time periods over which studies followed up? Skills and knowledge: are 5 studies referred to in this section, rather than the 4 reported at the start of this section? Page 12: not clear where the evidence is to support the conclusion on ‘most evidence’ for multifaceted studies. This is different than saying multifaceted interventions are likely to be successful based on the findings from the review, as stated in the following paragraph.
--	--

REVIEWER	Alison Cowley Nottingham University Hospitals NHS Trust, Research & Innovation
REVIEW RETURNED	25-Feb-2022

GENERAL COMMENTS	Dear Authors, I thought this was a clear and well written paper with strong methodology and reporting of results. The topic area is highly relevant and merits publication. There are a few minor points that need clarifying before accepting this article for publication. I have outlined these by section for ease of response. Introduction - Felt very superficial. Reference what a clinical academic career is as there is a huge amount of variation in the roles. This may also be reflected in the differing results from different countries.
---

	 - What are the benefits of clinical academics from an individual, organisational and patient point of view? - More detail on gender and ethnicity – what is known about these disparities to date? Methods  - I assume the search terms are detailed in the protocol but can you include as an appendix? Or tabulate in the text? - Built in machine learning based prioritization I am not familiar with this software but how is this comparable to hand screening? I think readers need a little more detail on this PPI  - seems like a token sentence. What was their contribution to this study. More detail is required of their role and contribution . Results  - Why were the 101 other studies not synthesized further? More details as the rationale for this decision was unclear. - Synthesized quantitative data – how were these categories decided? Were they a priori or did they emerge from the findings? Please ensure this matches during the methods section of data analysis – This section was really clear and easy to read. Discussion  - Greater detail on the use of machine learning – reference this and the methodological limitations and strengths of such an approach - So basically your study just confirmed what we already knew- I am unclear what new knowledge this study brings? I know this was partly covered in the introduction - Limited use of referencing in this section when comparing and contrasting evidence. This section could be strengthened by considering evidence wider than medical and dental clinical academic careers. For example, nursing, midwifery and AHP research into clinical academic careers would provide a useful comparison and help explain some of your findings.
--	---

VERSION 1 – AUTHOR RESPONSE

Reviewer: 1 - Dr. Sue Latter, University of Southampton

Comments to the Author:

On the whole this is a well conducted review with clearly presented results and subsequent discussion and implications that follow from these. The critical details missing are on the details of the search process, and also the justification for and details about the decision to exclude 101 of the 148 studies in the interests of a feasible review.

Thank you for your considered critique. We hope that the additional information which is now included, helps to further describe this research.

Specific comments:

Abstract:

Objectives: is it 'a focus on addressing inequalities'? This did not seem the prime focus of the review. Perhaps better as 'including a focus on addressing inequalities'.

Thank you for this very helpful edit – we have taken your advice.

Is there a qualitative equivalent to 'risk of bias' that was applied, and therefore should be included in the Abstract, for balance?

We have amended the wording here to reflect quality assessment, which we believe can be applied to quantitative, qualitative and mixed methodologies. Full details of the tools used are provided in the main text (due to the number of tools used, it is not possible to name them all within the abstract).

Results: What is the 'general overview'? This is unclear, and is not reported in the body of the text either.

Thank you for pointing out this inconsistency, we have corrected it within the abstract.

Strengths and limitations:

'Identify...earlier' is a little unclear. Earlier than what?

The use of the prioritisation algorithm brings the most relevant records to the “top” of the list for screening, thus identifying relevant records earlier than with traditional methods. We have amended this sentence to reflect this.

Introduction:

Second paragraph: could the relevance of this be made more explicit?

We have clarified this further within the revised manuscript.

Interventions designed to increase career progression is stated as an aim – if the latter distinct from recruitment and retention, should it be included in the Abstract and the body of the text as a focus?

Thank you. We have generally included the concept of career progression within recruitment and retention, but for clarity have rephased the objectives within the abstract.

Is it 'a key group' or group of people plural wrt inequalities?

We have removed the word key from this sentence, and hope that this has resolved any lack of clarity.

Methods

It would be helpful to spell out in full which database the review was registered with.

We apologise for this omission, we had included the full registry name in the abstract, and have now also expanded here.

The search strategy seems somewhat unconventional and the authors says the details will be published separately, but sufficient details should be included in this paper.

We have expanded the “search and information sources” and “study selection and data extraction” sections of the methods to make the linked search and screening processes clearer.

No inclusion / exclusion criteria related to inequalities? Or career progression?

This was an intentional research design decision. Our aims were to identify any intervention to support clinical academics, and thus included all studies rather than privileging those which explore inequalities. Within our data extraction and synthesis, we then sought to identify any impact of interventions on the inequalities which are known to exist in clinical academia. Our inclusion criteria did relate to career progression: “We included quantitative and qualitative studies evaluating interventions to increase recruitment to, and improve retention in, clinical academic careers, using study-defined outcome measures relating to rates of joining or continuing within clinical academia”.

Data are plural.

Thank you for highlighting this error – we have corrected it.

Why was the inclusion date changed from 2004 to 2005? Assume this was to allow for the time lag between Athena Swan introduction in 2004 and any research conducted in its wake? Please clarify.

Yes, this was the rationale and we have added a clause accordingly.

The analysis plan was adjusted? Further detail and a rationale for this are required.

Given the very large volume of potentially relevant studies identified during the screening process, the analysis plan was adjusted to enable time and resource efficient reporting, in discussion with the project funders. As described below, a decision was made to only synthesise quantitative studies including a control group and studies using qualitative methods. We have moved the sentence relating to this from the study selection section of the manuscript to the paragraph detailing these changes in order to make this rationale clearer.

Study selection:

The rationale for excluding 101 studies needs to be included.

These 101 studies were quantitative studies not including a control group. These were not further synthesised due to the extensive number of studies identified. As such, a decision was made to synthesise only those studies most likely to contribute to answering the specific aims of the research. This was agreed to be studies using quantitative methods which included a control group and studies using qualitative methods to evaluate the interventions. The methodological rigour of controlled studies over uncontrolled studies also played a role in our decision making. We have made adjustments to the methods and study selection section of the results in order to make this clearer. It would be a considerable undertaking (with additional resources and funding) to analyse these additional studies, which we would be willing to collaborate with/support other researchers to perform.

Page 6: trainees were included or not included? Exclusion criteria states students were not included – if these are different than ‘trainees’ a word of explanation is needed.

Within the UK medical and dental workforce structure, “students” is used to refer to those who are completing their medical or dental degree (usually as undergraduates). “Trainees” are those who have completed their degree but undergo further post-graduate training as qualified doctors or dentists. Thus, studies including trainees were eligible to be included in the review, whilst those including only students were not. We have added a sentence to the inclusion and exclusion criteria section to address this issue.

It would be helpful to summarise the range and types of interventions included. Were interventions included if they promoted careers generally as opposed to CA careers specifically?

To be included within the review, the studies were required to assess interventions to promote clinical academic careers, rather than careers generally (see inclusion and exclusion criteria section). The range and types of interventions are outlined within the third paragraph of the study characteristics section on page 5 of the manuscript and in further detail within the results when appropriate. Additional information relating to the interventions is included in Appendix 1 and within the full report reference in the manuscript. Given the manuscript word count and clear directions for readers as to where to find this additional information, we have not expanded this data further.

It would be helpful to summarise the range and types of design of studies.

We apologise for the omission. A sentence describing the study designs has been added to each study characteristics section.

Can the authors comment on the time periods over which studies followed up?

This data was not extracted from the studies and therefore we are unable to accurately comment.

Skills and knowledge: are 5 studies referred to in this section, rather than the 4 reported at the start of this section?

Four studies (references 17, 29, 41, and 45) are described throughout this section.

Page 12: not clear where the evidence is to support the conclusion on 'most evidence' for multifaceted studies. This is different than saying multifaceted interventions are likely to be successful based on the findings from the review, as stated in the following paragraph.

We have amended the sentence including "most evidence" and hope that the intended meaning is now clearer.

Reviewer: 2 - Dr. Alison Cowley, Nottingham University Hospitals NHS Trust

Comments to the Author:

Dear Authors,

I thought this was a clear and well written paper with strong methodology and reporting of results. The topic area is highly relevant and merits publication.

There are a few minor points that need clarifying before accepting this article for publication. I have outlined these by section for ease of response.

Thank you for your kind words, we appreciate your review and hope we have addressed the points for clarification to your satisfaction.

Introduction

- *Felt very superficial. Reference what a clinical academic career is as there is a huge amount of variation in the roles. This may also be reflected in the differing results from different countries.*

In an attempt to keep the manuscript succinct and to highlight only key introductory information, we have perhaps been too brief. In an attempt to rectify this, without dedicating the entire manuscript to a complex exploration of all relevant issues, we have included a number of revisions within the introduction to capture the wide variety of clinical academic roles, the influences upon these, alongside additional background information.

- *What are the benefits of clinical academics from an individual, organisational and patient point of view?*

We have added a sentence to the introduction to address this comment.

- *More detail on gender and ethnicity – what is known about these disparities to date?*

We have added a further sentence to this paragraph expanding on this issue.

Methods

- *I assume the search terms are detailed in the protocol but can you include as an appendix? Or tabulate in the text?*

The Medline search strategy is now included as Appendix 1.

- *Built in machine learning based prioritization I am not familiar with this software but how is this comparable to hand screening? I think readers need a little more detail on this*

We have provided further detail within this section of the manuscript. In short, Rayyan prioritises records as possible includes based on what has previously been included (it "learns" from the reviewers), and was then used to facilitate the large volume of title and abstract screening.

PPI

- *seems like a token sentence. What was their contribution to this study. More detail is required of their role and contribution .*

We have expanded this sentence to include further detail of the public involvement.

Results

- *Why were the 101 other studies not synthesized further? More details as the rationale for this decision was unclear.*

As per our response to reviewer 1, these were quantitative studies not including a control group, which were not further synthesised due to the volume of data identified and their limited ability to address the research question.

- Synthesized quantitative data – how were these categories decided? Were they a priori or did they emerge from the findings? Please ensure this matches during the methods section of data analysis

These categories were identified following the extraction of all relevant reported outcomes, which then were grouped into the most relevant categories. We have clarified this further within the initially paragraph of the Synthesis of quantitative data section. The methods section is accurate and has not been expanded in the interests of brevity.

– This section was really clear and easy to read.

Thank you for this feedback.

Discussion

- Greater detail on the use of machine learning – reference this and the methodological limitations and strengths of such an approach

We have provided further details within the methods and elaborated further on the strengths and limitations within our discussion.

- So basically your study just confirmed what we already knew- I am unclear what new knowledge this study brings? I know this was partly covered in the introduction

We have added additional sentences within the discussion to address this concern.

- Limited use of referencing in this section when comparing and contrasting evidence. This section could be strengthened by considering evidence wider than medical and dental clinical academic careers. For example, nursing, midwifery and AHP research into clinical academic careers would provide a useful comparison and help explain some of your findings.

Given the clear focus of our review on careers of doctors and dentists, we feel that it would not be appropriate to refer to evidence relating to very different populations and professions (including nursing, midwifery and AHPs), particularly given that the research into these groups frequently outlines how they are distinctly different from the medical/dental experience we describe. We would be strongly supportive of future reviews addressing the issues relating to inequalities in other healthcare professions.

VERSION 2 – REVIEW

REVIEWER	Sue Latter University of Southampton
REVIEW RETURNED	11-Jul-2022
GENERAL COMMENTS	Dear Authors I have reviewed all responses and am satisfied that these address my initial comments, making this valuable review ready for publication in my opinion.